# A prebiotic diet modulates microglial states and motor deficits in α-synuclein overexpressing mice

**Reem Abdel-Haq[1,2], Johannes CM Schlachetzki[3], Joseph C Boktor[1], Thaisa M Cantu-Jungles[4], Taren Thron[1], Mengying Zhang[1], John W Bostick[1], Tahmineh Khazaei[1], Sujatha Chilakala[5], Livia H Morais[1], Greg Humphrey[6], Ali Keshavarzian[7,8], Jonathan E Katz[5], Matthew Thomson[1], Rob Knight[6,9,10,11], Viviana Gradinaru[1,2], Bruce R Hamaker[4], Christopher K Glass[3], Sarkis K Mazmanian[1,2]***

[1]Division of Biology and Biological Engineering, California Institute of Technology, Pasadena, United States; [2]Aligning Science Across Parkinson's (ASAP) Collaborative Research Network, Chevy Chase, United States; [3]Department of Cellular and Molecular Medicine, University of California, San Diego, San Diego, United States; [4]Department of Food Science, Whistler Center for Carbohydrate Research, Purdue University West Lafayette, West Lafayette, United States; [5]Lawrence J Ellison Institute for Transformative Medicine, University of Southern California, Los Angeles, United States; [6]Department of Pediatrics, University of California, San Diego, San Diego, United States; [7]Department of Internal Medicine, Division of Gastroenterology, Rush University Medical Center, Chicago, United States; [8]Rush Center for Integrated Microbiome and Chronobiology Research, Rush University Medical Center, Chicago, United States; [9]Department of Computer Science and Engineering, University of California, San Diego, San Diego, United States; [10]Department of Bioengineering, University of California, San Diego, San Diego, United States; [11]Center for Microbiome Innovation, University of California San Diego, San Diego, United States

*For correspondence:
sarkis@caltech.edu

**Abstract** Parkinson's disease (PD) is a movement disorder characterized by neuroinflammation, α-synuclein pathology, and neurodegeneration. Most cases of PD are non-hereditary, suggesting a strong role for environmental factors, and it has been speculated that disease may originate in peripheral tissues such as the gastrointestinal (GI) tract before affecting the brain. The gut microbiome is altered in PD and may impact motor and GI symptoms as indicated by animal studies, although mechanisms of gut-brain interactions remain incompletely defined. Intestinal bacteria ferment dietary fibers into short-chain fatty acids, with fecal levels of these molecules differing between PD and healthy controls and in mouse models. Among other effects, dietary microbial metabolites can modulate activation of microglia, brain-resident immune cells implicated in PD. We therefore investigated whether a fiber-rich diet influences microglial function in α-synuclein overexpressing (ASO) mice, a preclinical model with PD-like symptoms and pathology. Feeding a prebiotic high-fiber diet attenuates motor deficits and reduces α-synuclein aggregation in the substantia nigra of mice. Concomitantly, the gut microbiome of ASO mice adopts a profile correlated with health upon prebiotic treatment, which also reduces microglial activation. Single-cell RNA-seq analysis of microglia from the substantia nigra and striatum uncovers increased pro-inflammatory signaling and reduced homeostatic responses in ASO mice compared to wild-type counterparts on standard diets. However, prebiotic feeding reverses pathogenic microglial states in ASO mice and promotes expansion of protective disease-associated macrophage (DAM) subsets of microglia. Notably, depletion of

microglia using a CSF1R inhibitor eliminates the beneficial effects of prebiotics by restoring motor deficits to ASO mice despite feeding a prebiotic diet. These studies uncover a novel microglia-dependent interaction between diet and motor symptoms in mice, findings that may have implications for neuroinflammation and PD.

## Editor's evaluation

What should Parkinson's Disease patients eat? This study shows that dietary fiber impacts gut microbes and immune cells in the brain of a mouse model of Parkinson's. These findings will enable follow-up studies aimed at figuring out how this works and efforts to translate these findings to improve patient care.

## Introduction

Parkinson's disease (PD) is the second most common neurodegenerative disorder in the United States and affects ~1% of the population over the age of 65. The incidence rate of PD is projected to double between 2015 and 2040, mainly due to lifestyle factors and increased lifespan (*Dorsey et al., 2018*). Clinical features of PD include slowed movement, muscle rigidity, resting tremors, and postural instability. These symptoms result from death of dopaminergic neurons of the nigrostriatal pathway regulating motor function (*Poewe et al., 2017*). Abnormal aggregation of the neuronal protein α-synuclein (αSyn) promotes disruptions in multiple cellular processes that contribute to neurodegeneration, including mitochondrial dysfunction, oxidative stress, proteasomal impairment, autophagy deficits, and neuroinflammation (*Poewe et al., 2017*).

Although PD is predominantly classified as a brain disorder, 70–80% of patients experience gastrointestinal (GI) symptoms, mainly constipation but also abdominal pain and increased intestinal permeability that usually manifests in the prodromal stages (*Forsyth et al., 2011*; *Yang et al., 2019*). Braak's hypothesis postulated nearly 20 years ago that αSyn aggregation may start at peripheral environmental interfaces, like the GI tract or olfactory bulb, and eventually reach the brain stem, substantia nigra, and neocortex via the vagus nerve (*Braak et al., 2003*). Increasing evidence has corroborated the potential for gut-to-brain spread of αSyn pathology in rodents (*Kim et al., 2019*; *Liu et al., 2017a*; *Svensson et al., 2015*). Additionally, several studies have detected differences in gut microbiome composition between PD patients and healthy controls (*Çamcı and Oğuz, 2016*; *Keshavarzian et al., 2015*; *Scheperjans et al., 2015*; *Tan et al., 2014*), with decreased abundance of health-promoting bacteria and an increase in pro-inflammatory pathogenic bacteria in the PD microbiome. Altering the microbiome in α-synuclein overexpressing (ASO) mice modulates brain pathology and motor performance (*Sampson et al., 2016*), and gut bacterial species have been shown to accelerate disease in other PD mouse models (*Choi et al., 2018*; *Sampson et al., 2020*). Additionally, antibiotic treatment improves motor symptoms in several models of PD (*Cui et al., 2022*; *Pu et al., 2019*; *Sampson et al., 2016*).

One potential target of gut-brain signaling in PD are microglia, a versatile macrophage-like population of brain cells that can shape neural circuity through regulation of neurogenesis, synaptic pruning, and myelination (*Anderson and Vetter, 2019*). In PD and other neurodegenerative conditions, microglial cellular repair responses are thought to become dysregulated, ultimately resulting in heightened reactivity and chronic inflammation that drives neurodegeneration (*Troncoso-Escudero et al., 2018*). Microglia respond to signals from within the brain, but also receive input from the periphery including from the gut microbiome (*Abdel-Haq et al., 2019*). Offspring of germ-free (GF) mice show differences in microglial gene expression and chromatin accessibility compared to specific-pathogen-free (SPF) counterparts (*Thion et al., 2018*). Microglia from adult GF mice present an immature gene expression profile and fail to adequately respond to immunostimulants (*Erny et al., 2015*; *Thion et al., 2018*). However, feeding GF mice a mixture of short-chain fatty acids (SCFAs), metabolic products of bacterial fiber fermentation, is sufficient to rescue microglial maturation (*Erny et al., 2015*). Interestingly, levels of SCFAs are reduced in fecal samples from PD patients compared to matched controls (*Chen et al., 2022*; *Unger et al., 2016*) and inversely correlate with disease severity (*Aho et al., 2021*; *Chen et al., 2022*).

Herein, we explore the interplay between diet and microglia in the ASO mouse model, which recapitulates many of the hallmark symptoms of PD including motor deficits, GI abnormalities, olfactory dysfunction, and neuroinflammation (*Chesselet et al., 2012*). We demonstrate that a prebiotic diet remodels the gut microbiome of ASO mice to contain increased relative abundances of taxa linked to outcomes associated with health. Prebiotic intervention attenuates motor deficits and reduces αSyn aggregates in the substantia nigra of ASO mice in a microglia-dependent manner. Prebiotic diet alters the morphology and gene expression patterns of microglia in brain regions involved in PD, promoting phenotypes associated with disease-protective responses. Importantly, microglial depletion abrogates the beneficial effects of prebiotics. Overall, this study reveals that enhanced metabolism of dietary fiber by the gut microbiome alters the physiology of cells in the CNS and ultimately improves behavioral and pathologic outcomes in a mouse model of PD.

## Results

### Prebiotic diet attenuates motor symptoms and reduces αSyn aggregation in the brain

We generated three custom high-fiber diets (*Supplementary file 1*), each containing 20% of a prebiotic mixture of two or three dietary fibers designed to promote growth of distinct gut bacterial taxa (*Figure 1—figure supplement 1A*) and boost SCFA production (*Figure 1—figure supplement 1B-E*) based on in vitro fecal fermentation. The prebiotic diets (*Figure 1—figure supplement 1F*) were compared to a cellulose-free control diet that is similar in major micro- and macro-nutrients (*Supplementary file 1*).

We fed each of the three prebiotic diets (prebiotic #1, #2, #3) to male ASO mice from 5 to 22 weeks of age. To assess whether long-term prebiotic intervention ameliorated motor deficits, mice were subjected to a battery of motor tests to evaluate fine motor control, grip strength, locomotion, and coordination (*Figure 1A–D*, *Figure 1—figure supplement 2A-G*). We identified a single prebiotic (prebiotic #1, referred to hereafter as 'prebiotic') that improved disease symptoms in ASO mice. Remarkably, administration of the prebiotic diet to ASO mice enhanced performance in several motor behavioral tests, including the pole descent and beam traversal tests (time to cross, steps to cross, errors per step) compared to mice fed a control diet (*Figure 1A–D*). Outcomes in other paradigms including adhesive removal, wire hang, and hindlimb score were unchanged (*Figure 1—figure supplement 3A-C*). These findings reveal that a gut-targeting intervention has the potential to attenuate key behavioral features in a mouse model of PD.

As anticipated, levels of all major SCFAs were higher in fecal samples from prebiotic-fed mice than from control-fed mice (*Figure 1E*). Concentrations of propionate, butyrate, and isobutyrate were not significantly different between wild type (WT) and ASO mice fed a control diet (*Figure 1E*). ASO mice weighed significantly less than their WT counterparts and exhibited reduced food intake of control diet, but not prebiotic diet (*Figure 1—figure supplement 3D-E*). While prebiotic-ASO mice ate significantly more than control-ASO mice, no difference in body weight was detected between the groups at 22 weeks of age (*Figure 1—figure supplement 3D*). There were no obvious health issues in animals on either diet.

Aggregation of αSyn is a hallmark of PD pathology. We found a significant reduction in αSyn aggregation in the substantia nigra (SN) of prebiotic-fed ASO mice compared to ASO mice on control chow (*Figure 1F*). In contrast, prebiotic intervention had no effect on αSyn aggregation in the striatum (STR) (*Figure 1G*). We speculate that this difference may be attributable to regional differences in microglia density, gene expression, and clearance activity (*Grabert et al., 2016*; Y.-L. *Tan et al., 2020*). Taken together, these results suggest that early intervention with a prebiotic diet can reduce PD-like symptoms and brain pathology in ASO mice.

### Prebiotics alter gut microbiome composition

Gut microbiome composition is strongly influenced by diet in mice and humans (*Turnbaugh et al., 2009*; *Wu et al., 2011*). We performed shotgun metagenomics on fecal samples from mice fed control or prebiotic diet. Alpha diversity analysis revealed significant reduction in observed species count, Shannon's diversity, and Simpson's evenness in prebiotic-fed groups, as well as an increase in Gini's dominance (*Figure 2A–D*). This is consistent with a previous report of reduced microbiome

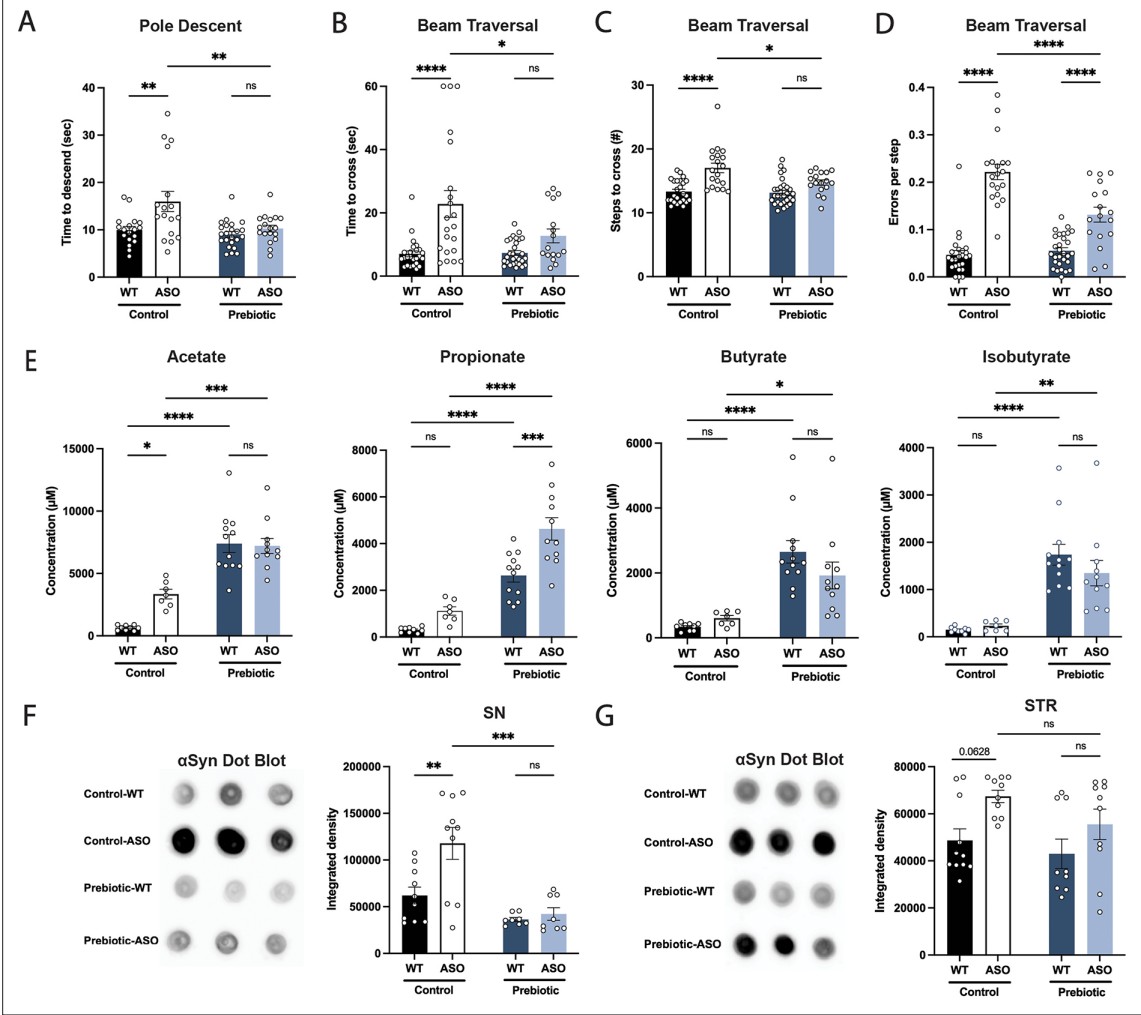

**Figure 1.** Prebiotic diet #1 attenuates motor symptoms and reduces αSyn aggregation. (**A–D**) Motor behavior metrics at 22 weeks of age for prebiotic- and control-fed WT and ASO mice from pole descent (**A**) and beam traversal (**B–D**) tests. Motor test data is derived from two independent experiments (n=16–29/group). (**E**) Concentrations (µM) of acetate, propionate, butyrate, and isobutyrate in fecal samples collected from prebiotic-fed WT and ASO mice (n=7–12/group). (**F–G**) Aggregated α-synuclein levels in the substantia nigra (SN) (F; n=8–10/group) and striatum (STR) (G; n=9–11/group) measured by dot blot. Each point represents data from one mouse. Data analyzed by two-way ANOVA followed by Tukey's multiple comparisons test. Bars represent mean ± SEM. *p<0.05, **p<0.01, ***p<0.001, and ****p<0.0001.

The online version of this article includes the following source data and figure supplement(s) for figure 1:

**Source data 1.** Original image of αSyn dot blot shown in **Figure 1F**.

**Source data 2.** Original image of αSyn dot blot shown in **Figure 1G**.

**Figure supplement 1.** Effect of dietary fibers on gut microbial community and metabolic function in vitro.

**Figure supplement 2.** Motor behavior in mice fed Prebiotic #2 and Prebiotic #3 diets.

**Figure supplement 3.** Prebiotic diet does not improve performance in certain motor tests.

diversity in high-fiber fed mice (**Luo et al., 2017**). Principal coordinate analysis (PCoA) of species abundance showed that samples clustered more closely by diet than mouse genotype (**Figure 2E**) and PERMANOVA revealed that prebiotic treatment explained 6-fold more variance than genotype, with $R^2$ values of 0.334 and 0.053 for each, respectively (**Figure 2F**). Thus, the prebiotic diet reshapes gut microbial communities in WT and ASO mice.

We observed broad changes at the microbial phylum and genus levels following administration of prebiotic diet (**Figure 2G and I**), displaying an increase in Bacteroidetes and a decrease in Firmicutes in prebiotic diet-fed mice, resulting in a lower Firmicutes/Bacteroidetes (F/B) ratio that has been associated with general features of metabolic health (**Figure 2H**). Intriguingly, it has been shown that

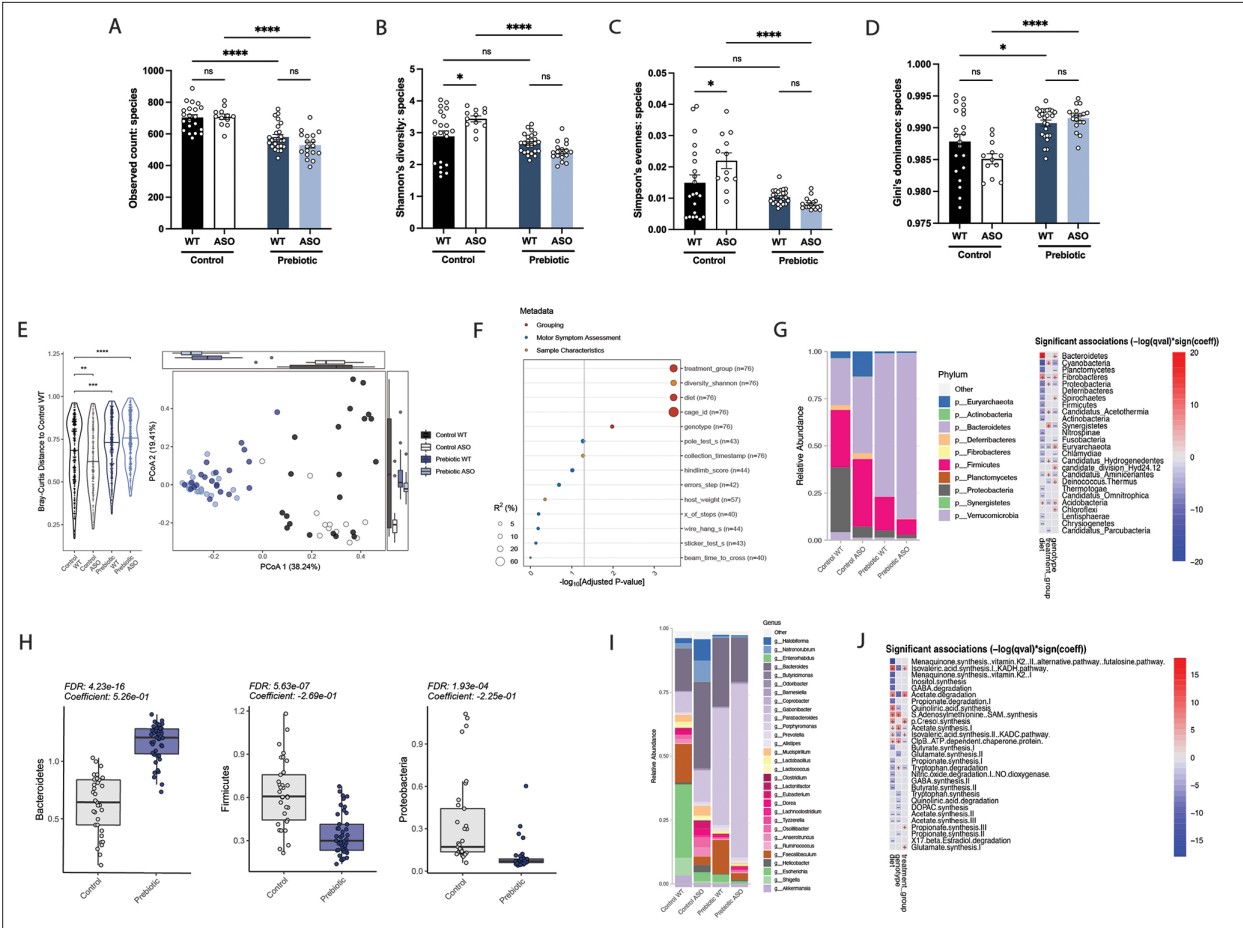

**Figure 2.** Mice fed a prebiotic diet display a distinctive gut microbiome compared to controls. (**A–D**) Diversity metrics from metagenomic analysis of treatment groups at 22 weeks of age, including observed species count (**A**), Shannon's diversity (**B**), Simpson's evenness (**C**), and Gini's dominance (**D**). (**E**) Principal Coordinate Analysis (PCoA) plot of Bray-Curtis dissimilarity (n=12–25/group). (**F**) PERMANOVA analysis summary of Bray-Curtis dissimilarity. (**G**) Relative abundance of phyla among treatment groups (left) and heat map showing differentially abundant phyla (right). Diet values are displayed relative to control diet and genotype values relative to WT mice. (**H**) Relative abundance of select phyla in treatment groups. (**I**) Summary plot of relative abundance of genera. (**J**) Differentially expressed pathways identified from the 'Gut Microbiome-Brain module'. Diet values are displayed relative to control diet and genotype values relative to WT mice (n=12–25/group).

Bacteroidetes are reduced in PD patients compared to age-matched controls, suggesting the prebiotic may counter this effect (*Unger et al., 2016*). Additionally, we observed a decrease in Proteobacteria, a phylum often increased in dysbiosis and inflammation and elevated in PD patient fecal samples (*Figure 2H*; *Keshavarzian et al., 2015*; *Shin et al., 2015*). Gut-brain module analysis showed variation in metabolic pathways including SCFA degradation/synthesis in response to diet and genotype (*Figure 2J*). Overall, feeding of a prebiotic diet appears to qualitatively restructure the ASO microbiome toward increased relative abundances of taxa associated with potentially protective effects.

## Prebiotic diet alters microglia morphology in ASO mice

In ASO mice, microglia reactivity in response to αSyn overexpression appears at 4–5 weeks of age in the STR and at 20–24 weeks of age in the SN (*Watson et al., 2012*). SCFAs have been shown to influence the physiology of microglia in several contexts (*Colombo et al., 2021*; *Erny et al., 2015*; *Sadler et al., 2020*; *Erny et al., 2021*; *Sampson et al., 2016*). To explore whether prebiotics alter microglia morphology, we performed immunofluorescence imaging using the pan-microglial marker IBA1. The morphology of microglia can indicate their reactivity state, with homeostatic microglia exhibiting a ramified shape with a smaller cell body and increased dendritic processes, whereas activated microglia adopt an amoeboid form with a larger cell body and retracted processes (*Menassa and Gomez-Nicola, 2018*). We observed that microglia in the SN and STR of prebiotic-ASO mice had significantly

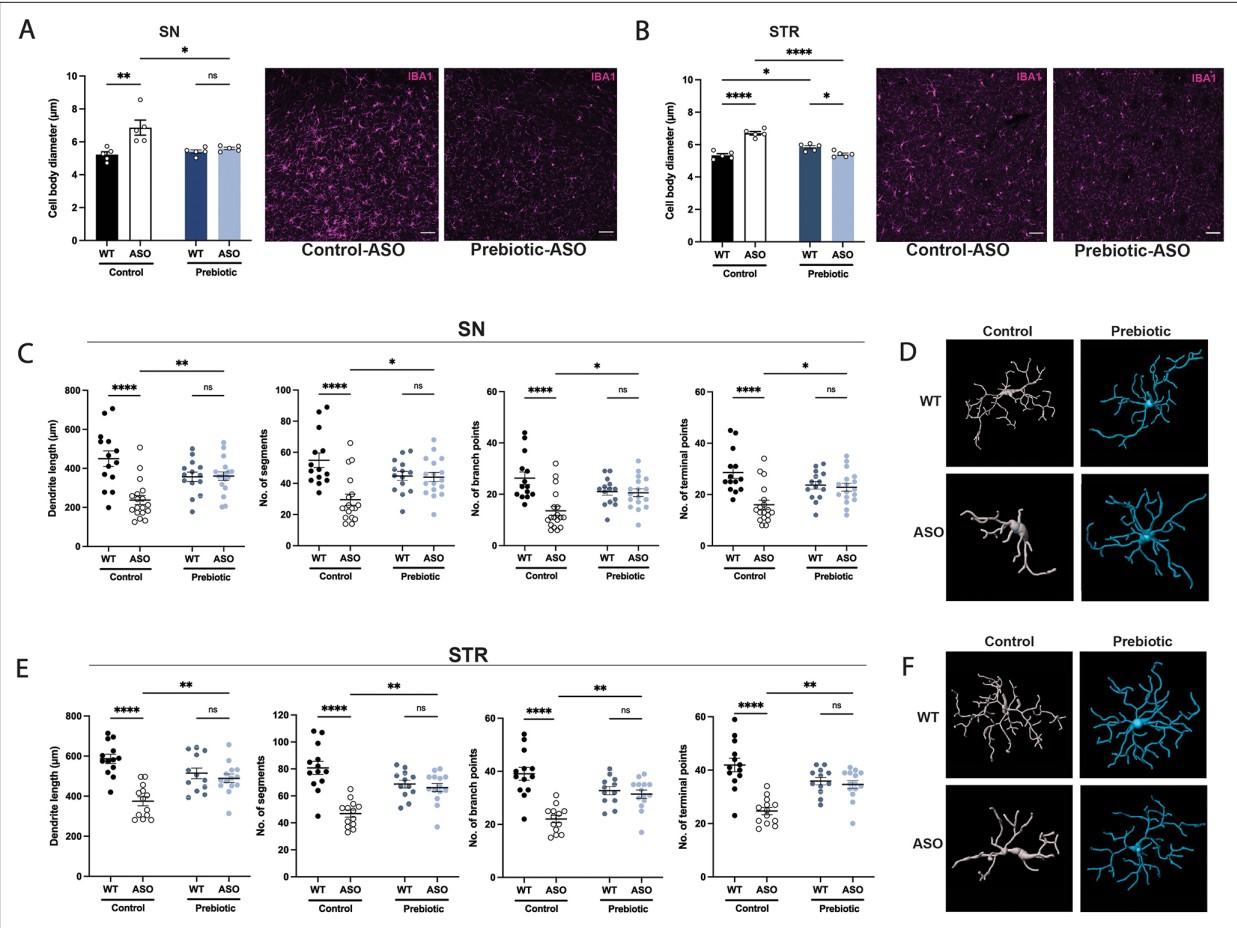

**Figure 3.** Prebiotic diet alters microglia morphology and reactivity status in ASO mice. (**A,B**) Measurement of IBA1+ microglia diameter in substantia nigra (SN) (A; n=5/group) and striatum (STR) (B; n=5/group). Left: quantification of cell diameter. Each point represents one mouse with 26–79 cells measured per mouse. Right: Representative 20 X images of IBA1 staining. Scale bars 50 µm. (**C–F**) 3D reconstruction of microglia in the substantia nigra (**C–D**) and striatum (**E–F**). (**C,E**) Quantification of dendrite length, number of segments, number of branch points, and number of terminal points (n=14–18/group for SN and n=12–14/group for STR). Each point represents one cell, with 3–5 cells analyzed/mouse. (**D,F**) Representative 3D reconstructions of microglia imaged at 40 X magnification. Data analyzed by two-way ANOVA followed by Tukey's multiple comparisons test. Bars represent mean ± SEM. *p<0.05, **p<0.01, ***p<0.001, and ****p<0.0001.

smaller cell bodies than in control-ASO mice (*Figure 3A–B*). 3D analysis of key morphological features revealed that microglia in the SN and STR of prebiotic-ASO mice exhibited increased dendrite length, number of segments, number of branch points, and number of terminal points compared to microglia from control-ASO mice (*Figure 3C–F*). Taken together, these findings indicate that long-term prebiotic intervention dampens microglial reactivity in brain regions implicated in PD.

## ASO mice display increased disease-promoting microglial subsets

Single-cell RNA sequencing (scRNA-seq) has emerged as a powerful tool to interrogate microglial biology in mouse models of neurodegeneration (*Keren-Shaul et al., 2017*; *Liu et al., 2020*). We first sought to investigate differences in microglial gene expression between control-WT and control-ASO mice (no prebiotics), since scRNA-seq of microglia has not been previously applied to this mouse model. Differential gene expression analysis of all cells revealed 313 differentially expressed genes (DEGs) (↑163, ↓150, FDR <0.05) in the SN and 997 DEGs (↑511, ↓486) in the STR. In the SN, microglia harvested from control-ASO mice displayed increased expression of MHC class I components (*H2-K1*, *H2-D1*), several chemokines (*Ccl2, Ccl3, Ccl4, Ccl9*) and chemokine receptors (*Ccr1, Ccr5*), and proinflammatory markers (*Nfkbid, Cd 74*) (*Figure 4C*, *Supplementary file 2a*). Gene enrichment analysis of all upregulated DEGs in control-ASO mice showed enrichment in pathways related to cellular responses to cytokine stimulus and interferon-gamma, immune system processes, and response

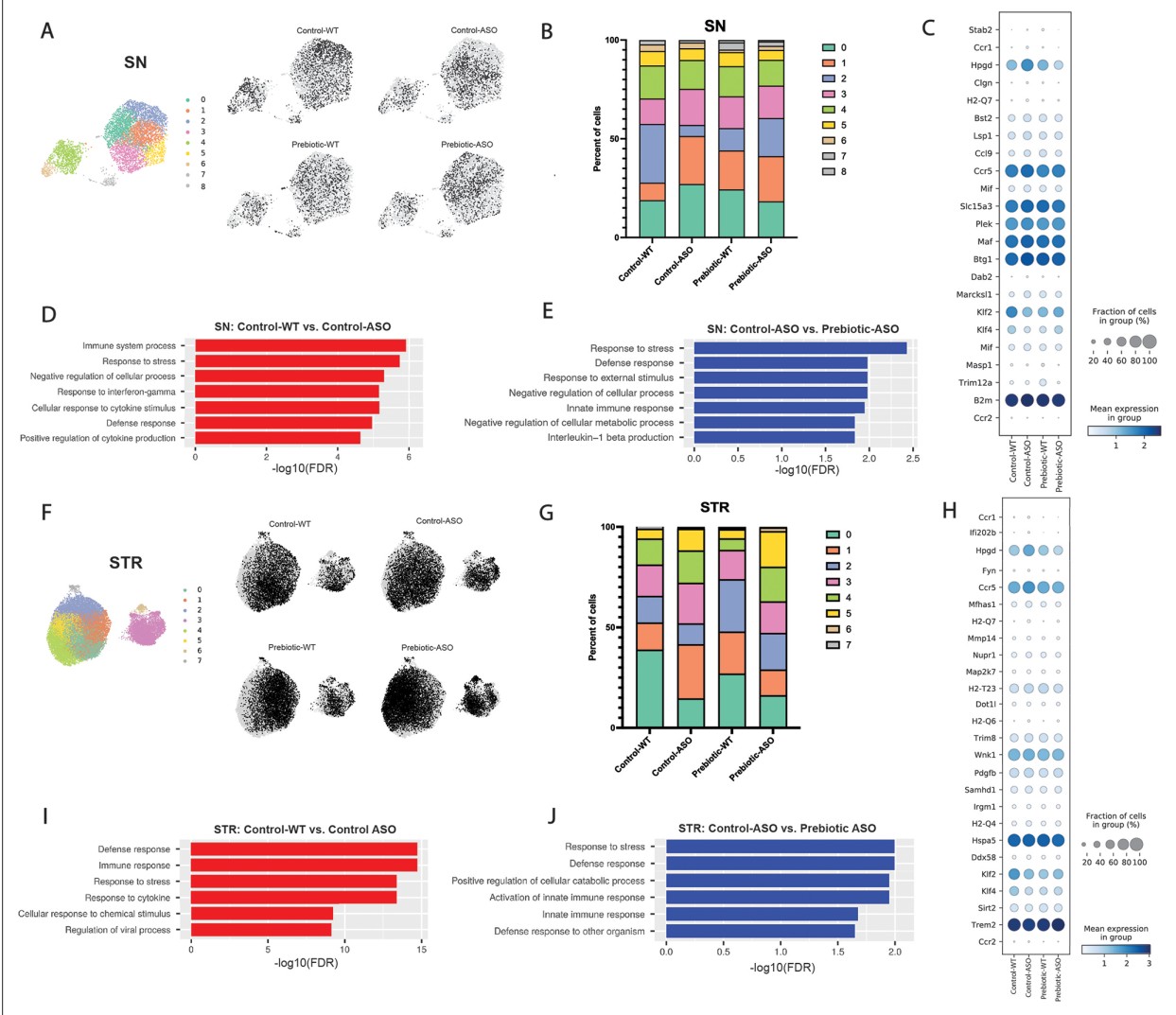

**Figure 4.** Prebiotic diet alters microglial gene expression. (**A**) UMAP plot of all 5278 substantia nigra (SN) cells sequenced by scRNA-seq from all four treatment groups (left) and distribution of cells from individual samples (right). (**B**) Relative distribution of cells within each cluster in the SN. (**C**) Dot plot showing genes significantly upregulated in control-ASO microglia (relative to control-WT) and significantly downregulated in prebiotic-ASO microglia (relative to control-ASO) in the SN. (**D**) Significantly enriched pathways among 163 genes upregulated in control-ASO microglia relative to control-WT microglia in the SN. (**E**) Significantly enriched pathways among 156 genes downregulated in prebiotic-ASO microglia relative to control-ASO microglia in the SN. (**F**) UMAP plot of all 27,152 striatal (STR) cells sequenced by scRNA-seq from all four treatment groups (left) and distribution of cells from individual samples (right). (**G**) Relative distribution of cells within each cluster in the STR. (**H**) Dot plot and showing genes significantly upregulated in control-ASO microglia (relative to control-WT) and significantly downregulated in prebiotic-ASO microglia (relative to control-ASO) in the STR. (**I**) Significantly enriched pathways among the 50 most upregulated genes in control-ASO microglia relative to control-WT microglia in the STR. (**J**) Significantly enriched pathways among the 50 most downregulated genes in prebiotic-ASO microglia relative to control-ASO microglia in the STR.

The online version of this article includes the following figure supplement(s) for figure 4:

**Figure supplement 1.** Prebiotics do not change SCFA levels in the brain.

**Figure supplement 2.** FFAR2/3 levels in brain and GI tissue and epigenetic analysis.

to stress (*Figure 4D*). Several genes that were downregulated in control-ASO mice compared to control-WT were related to anti-inflammatory signaling (*Klf2*, *Klf4*) and microglial homeostasis (*P2ry12*, *Slc2a5*) (*Figure 4C*, *Supplementary file 2a*). We observed similar trends in the STR, with control-ASO microglia upregulating pro-inflammatory modulators (*Tnf*, *Nfkbiz*, *Trim8*, *Irgm1*) and antigen processing and presentation genes (*H2-Q7*, *H2-K1*, *H2-D1*, *H2-T23*) and downregulating genes related to homeostatic cellular processes (*Figure 4H–I*, *Supplementary file 2c*). Notably, the anti-inflammatory cytokine transforming growth factor beta 2 (*Tgfβ2*) was ~45-fold downregulated in

control-ASO (*Supplementary file 2c*). These data suggest microglia from control-ASO mice upregulate pro-inflammatory immune processes and downregulate pathways related to homeostasis and cellular maintenance in response to αSyn overexpression.

## Prebiotic diet promotes microglia with disease-protective functions

Based on global scRNA-seq gene expression, Uniform Manifold Approximation and Projection for Dimension Reduction (UMAP) analysis yielded nine distinct microglia clusters in the SN and eight clusters in the STR (*Figure 4A and F*). In the SN, we detected differences in cluster distribution between experimental groups, with the strongest differences in clusters 0 and 2 (*Figure 4A–B*). Interestingly, the percentage of microglia in cluster 0 was higher in control-ASO than control-WT mice (27.1% vs 18.9%), and prebiotic treatment reduced the percentage of microglia belonging to cluster 0 in ASO mice compared to control diets (18.3%) (*Figure 4B*). Gene enrichment analysis of the top 50 genes associated with cluster 0 revealed pathways related to immune system processes, cellular response to tumor necrosis factor (TNF), cellular response to lipopolysaccharide, and response to stress. Cluster 0 contained several prominent immune markers including *Tnf*, *Nfkbia*, *Ccl2*, *Ccl3*, and *Ccl4*, suggesting that a prebiotic diet may suppress or prevent pro-inflammatory responses in ASO mice. Notably, levels of TNF and Ccl2 are elevated in the serum of PD patients (*Brodacki et al., 2008*; *Reale et al., 2009*). Conversely, the percentage of microglia belonging to cluster 2 was reduced in control-ASO mice but increased in prebiotic-ASO mice (*Figure 4B*). Among the most highly expressed genes in cluster 2 were the homeostatic microglial markers *P2ry12* and *Cst3*, as well as the anti-inflammatory transcription factors *Klf2* and *Klf4* (*Das et al., 2006*; *Li et al., 2018*).

Within the STR, we detected eight clusters of microglia, with notable shifts in clusters 1 and 3 (*Figure 4F–G*). The top 10 associative genes in cluster 3 included several mitochondrial genes: *mt-Atp6*, *mt-Cytb*, *mt-Co2*, *mt-Co3*, *mt-Nd4*, *mt-Nd1*, and *mt-Nd2*. Additionally, we detected a 13.4% increase in cluster 1 in control-ASO mice, with prebiotic diet restoring the percentage of cluster 1 back to control-WT levels (*Figure 4G*). The significantly enriched pathways within cluster 1 included those positively regulating cell death and immune system development, and negatively regulating cellular processes, suggesting increased immune signaling and dysregulation of homeostatic signaling in the absence of prebiotic treatment.

To determine effects of long-term prebiotic exposure on microglial gene expression in ASO mice, we compared prebiotic-ASO microglia to control-ASO and found 473 DEGs (↑317, ↓156) in the SN and 1474 DEGs (↑608, ↓866) in the STR (*Figure 4C and H*, *Supplementary file 2b, d*). Gene enrichment analysis of the 156 genes downregulated in prebiotic-ASO microglia in the SN revealed reduction in interleukin-1 (IL-1)β production pathways, as well as dampened innate immune response and defense response pathways compared to control-ASO mice (*Figure 4E*). Among the genes downregulated in microglia from prebiotic-ASO mice were several mediators of the pro-inflammatory immune response (*Mif*, *Masp1*, *Trim12a*, *Bs2*, *B2m*), antigen presentation and processing (*H2-Q7*), and chemokines/receptors (*Ccl9*, *Ccr1*, *Ccr5*) (*Figure 4C*, *Supplementary file 2b*). We observed a similar trend in the STR, with prebiotic-ASO showing downregulation of pathways related to innate immunity, response to stress, and defense response (*Figure 4H and J*, *Supplementary file 2d*). Interestingly, several of the pro-inflammatory markers upregulated in control-ASO and downregulated in prebiotic-ASO microglia were expressed by a small subset of microglia, suggesting that a subpopulation of cells alters its transcriptomic profile in response to αSyn expression, similar to what has been observed in microglia from aged mice and a mouse model of Alzheimer's disease (AD) (*Hammond et al., 2019*; *Keren-Shaul et al., 2017*). Further DEG analysis revealed increased expression of several markers that define disease-associated macrophages (DAM) in the SN and STR in prebiotic-ASO mice (*Supplementary file 2b, d*), a microglial sub-population associated with protection during early stages of disease in several mouse models (*Deczkowska et al., 2018*; *Onuska, 2020*). Notably, we observed an increase in Trem2 in microglia from the STR of prebiotic-ASO mice, suggesting prebiotics may induce a neuroprotective DAM phenotype by 22 weeks of age (*Gratuze et al., 2018*; *Keren-Shaul et al., 2017*; *Onuska, 2020*). Taken together, gene expression analysis suggests prebiotic intervention in ASO mice dampens proinflammatory and neurotoxic signaling pathways and potentially upregulates a neuroprotective phenotype in microglia.

## Potential effects of SCFAs are likely indirect and not via epigenetic regulation

We detected no differences in SCFA levels between control and prebiotic animal groups in either the SN or STR (*Figure 4—figure supplement 1A-B*). SCFAs can signal through activation of GPCR receptors (GPCR43 or FFAR2, and GPCR41 or FFAR3) and/or inhibition of histone deacetylases (HDACs), altering the epigenetic landscape of target cells (*Silva et al., 2020*; *Vinolo et al., 2011*). As determined via qRT-PCR, ASO mice exhibited very low or no expression of FFAR2 and FFAR3 in the cerebellum, midbrain, striatum, and motor cortex relative to the small intestine (*Figure 4—figure supplement 2A, B*), consistent with scRNA-seq data showing an absence of FFAR2/3 expression in microglia in the SN and STR (*Supplementary file 3*).

To explore whether the prebiotic diet was inducing epigenetic changes, we performed bulk ATAC-seq on purified microglia from the SN and STR and observed no significant differences in chromatin accessibility between experimental groups (*Figure 4—figure supplement 2C, D*). However, from this bulk measurement, we cannot rule out changes in open chromatin or histone modifications in specific subset(s) of microglia. We also measured the expression levels of several HDAC isoforms (*Hdac 1, 2, 6, 7, and 9*) in the striatum and found no differences in expression between control and prebiotic groups of both genotypes (*Figure 4—figure supplement 2E-I*). Collectively, these findings suggest that dietary metabolites may influence microglial gene expression through indirect mechanisms and likely not by entering the brain, consistent with previous reports (*Erny et al., 2015*), though additional work is needed to validate this hypothesis.

## Depletion of microglia blocks beneficial effects of prebiotics

Microglia are dependent on colony stimulating factor 1 receptor (CSF1R) signaling for development, maintenance, and proliferation (*Elmore et al., 2014*). PLX5622 is a brain-penetrant inhibitor of CSF1R that can deplete microglia with no observed effects on behavior or cognition (*Elmore et al., 2014*). We added PLX5622 to the diet of mice from 5 to 22 weeks of age, and quantified the number of IBA1 +microglia in various brain regions. The efficiency of microglial depletion varied depending on brain region, with regions containing low numbers of microglia such as the cerebellum exhibiting higher depletion (~80%) than areas with high microglial density such as the SN (~65%) and STR (~75%) (*Figure 5A–C*). We did not observe differences in depletion efficiency between WT and ASO mice or between control and prebiotic-fed mice (*Figure 5—figure supplement 1A-B*).

Following PLX5622 treatment, we assayed motor behavior at 22 weeks of age. PLX5622 treatment had no impact on motor performance in tests where prebiotic treatment had no effect (*Figure 5—figure supplement 1C-F*). Remarkably however, even incomplete microglia depletion eliminated prebiotic-induced improvements in the pole descent and beam traversal tests (time to cross, errors per step) (*Figure 5D–F*), suggesting that microglia are required for the ability of prebiotics to ameliorate motor deficits. PLX5622 treatment did not alter body weight in control or prebiotic-fed mice (*Figure 5—figure supplement 1G*). We also measured αSyn aggregation in the SN and STR of 22-week-old mice. In control-fed mice, depletion of microglia had no impact on levels of αSyn aggregation in the SN or STR (*Figure 5G–H*). However, in prebiotic-fed WT and ASO mice, depletion of microglia significantly increased levels of aggregated αSyn in the SN, while levels in the STR remained unchanged (*Figure 5G–H*). These data reveal that partial ablation of microglia or diminished CSF1R signaling eliminate the protective effects of the prebiotic diet in ASO mice.

While previous studies have characterized the effect of PLX5622 on macrophages in the spleen and bone marrow (*Lei et al., 2020*), knowledge of the effect of this drug on immune cell populations in the GI tract of mice is largely unexplored. Surprisingly, most gut-associated immune cell populations were unaffected by PLX5622 treatment. In the large intestine, PLX5622 treatment caused a reduction in CD45+ CSF1Rlo lymphocytes, but had no impact on CD45+ CSF1Rhi cells, pan T cells or B cells (*Figure 5—figure supplement 2A-E*). In the small intestine, levels of these cell types were unchanged in response to PLX5622 (*Figure 5—figure supplement 2F-J*). In the spleen, while CSF1Rlo lymphocytes were reduced in Prebiotic +PLX5622 mice, levels of CSF1Rhi macrophages were significantly elevated in Control +PLX5622 and Prebiotic +PLX5622 mice, suggesting a potential compensatory mechanism in this organ (*Figure 5—figure supplement 2K-O*). These findings point to a relatively high specificity of CSF1R-targeted depletion in the brain, further implicating microglia as a key mediator of the beneficial effects of prebiotic treatment in ASO mice.

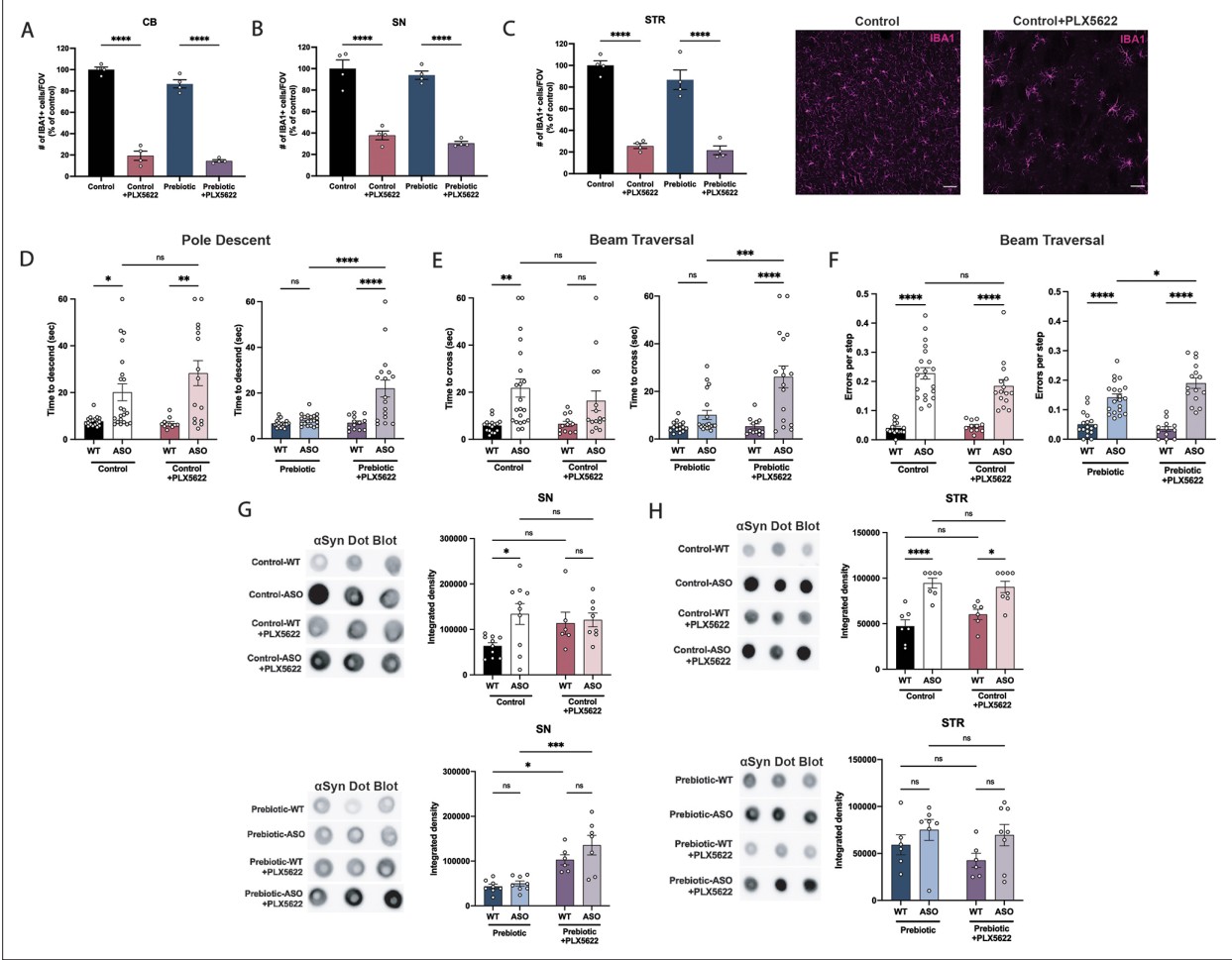

**Figure 5.** Depletion of microglia inhibits beneficial effects of prebiotics. (**A–C**) Number of IBA1+ cells per field of view in 20 X images of the cerebellum (**A**), substantia nigra (**B**), and striatum (**C**). n=4/group. Representative images from the striatum are shown at right (scale bars: 50 µm). (**D–F**) Motor performance metrics for pole descent (**D**) and beam traversal (**E–F**) tests. Motor data derived from five independent cohorts (n=12–21/group). (**G,H**) Aggregated α-synuclein measured by dot blot in the substantia nigra (G; n=6–10/group) and striatum (H; n=6–8/group). Microglia count data analyzed by one-way ANOVA followed by Tukey's multiple comparisons test. Motor and αSyn data analyzed by two-way ANOVA followed by Tukey's multiple comparisons test. Data represent mean ± SEM. *p<0.05, **p<0.01, ***p<0.001, and ****p<0.0001.

The online version of this article includes the following source data and figure supplement(s) for figure 5:

**Source data 1.** Original image of αSyn dot blot shown in *Figure 5G*.

**Source data 2.** Original image of αSyn dot blot shown in *Figure 5H*.

**Figure supplement 1.** Characterization of PLX5622 treatment.

**Figure supplement 2.** Immune cell characterization in the gut and spleen of PLX5622-treated mice.

## Discussion

We describe how administering a prebiotic diet to α-synuclein overexpressing mice results in improved motor performance with reduced microglial reactivity and αSyn pathology. The mechanism by which a high-fiber diet influences microglial physiology and alters behavior remains unclear. SCFA levels in the brain tissue of prebiotic-fed mice were unchanged, and our data suggest that SCFAs do not appear to signal through known GPCRs in the brain or via epigenetic remodeling of microglia-derived chromatin, further reinforcing the notion of indirect effects on microglia, as previously suggested (*Erny et al., 2015*). SCFAs are known to have immune modulatory properties in the gut (*Parada Venegas et al., 2019*), among other functions, and we speculate that altering peripheral immunity may affect microglial reactivity states and gene expression. We note that it is possible molecules other than

SCFAs may be contributing to prebiotic-induced changes in microglial physiology, a notion we are unable to test in the ASO mouse model.

Studies of SCFAs in preclinical models paint a complex picture, with varying outcomes in germ-free (GF) vs. SPF settings. Oral administration of SCFAs to GF mice induces microglial reactivity in wild-type mice (*Erny et al., 2015*), a mouse model of AD (*Colombo et al., 2021*), and ASO mice, where feeding the metabolites in the absence of gut bacteria exacerbates motor deficits and neuroinflammation (*Sampson et al., 2016*). In contrast, two independent studies found that sodium butyrate treatment alleviates motor deficits and reduces microglial reactivity in 1-methyl-4-phenyl-1,2,3,6-tetrahydropyridine (MPTP) mice with a laboratory microbiota (*Hou et al., 2021*; *Liu et al., 2017b*). Our findings underscore the need to consider context (GF vs. SPF), diet, and form and duration of intervention in future diet studies in mouse models.

Microglia have been increasingly linked to neurodegenerative disorders and PD. Depletion of microglia using CSF1R inhibitors confers deleterious effects in certain mouse models of PD (MPTP, human α-Syn AAV) (*George et al., 2019*; *Yang et al., 2018*), LPS-induced sickness behavior (*Vichaya et al., 2020*), and prion disease (*Carroll et al., 2018*). In contrast, microglia depletion improves disease outcome in experimental autoimmune encephalomyelitis (EAE), a preclinical model of multiple sclerosis (*Nissen et al., 2018*), and in the 3xTg and 5xFAD mouse models of AD (*Casali et al., 2020*; *Spangenberg et al., 2016*; *Spangenberg et al., 2019*). Herein, we found that depletion of microglia neither exacerbates nor improves motor performance in naïve (control diet) mice, suggesting that microglia do not influence behavior in ASO mice, at least in the early stages of disease progression. In contrast, the protective effects of a prebiotic diet do require microglia since their depletion eliminated improvements in motor behavior and αSyn pathology in the brain. Our study does not rule out indirect effects of PLX5622 that include reshaping the microbiome to promote motor symptoms in prebiotic diet-fed mice.

We extended these findings with scRNA sequencing, uncovering functional effects including prebiotic-mediated restoration of pathways known to be dysregulated in PD including inflammation and homeostatic cellular functions. Moreover, we found that prebiotic intervention significantly increases CSF1 expression in ASO microglia in both the SN and STR, potentially implicating CSF1 signaling pathways in mediating the protective effects of prebiotics. Further insights into how prebiotic diets modulate microglia biology and how these events translate into amelioration of motor symptoms and brain pathology await future research. Microglia have been shown to present a distinct transcriptomic profile and respond to various environmental factors, including the microbiome, in a sex-specific manner (*Thion et al., 2018*; *Villa et al., 2018*). While this study probed the effects of prebiotics on microglia in male mice, additional insight may come from similar investigation of female animals.

Prebiotics present a potentially promising therapeutic approach as diet is a significant contributor to microbiome composition and epidemiological evidence has linked high-fiber diets with reduced risk of developing PD (*Boulos et al., 2019*). While increased intake of fruits, vegetables, and adherence to a Mediterranean diet are associated with a lower risk of PD, individuals consuming a low-fiber, highly processed Western diet exhibit an increased risk of PD diagnosis (*Alcalay et al., 2012*; *Gao et al., 2007*; *Molsberry et al., 2020*). Several ongoing clinical trials are exploring the beneficial effects of probiotics and prebiotics on PD-related outcomes. Gut-targeted therapies offer several advantages compared to traditional therapeutic approaches for brain disorders. Conventional pharmacological treatments rely on chemicals which may lose efficacy over time, often fail to treat underlying pathophysiology, and may result in undesirable side effects for the patient. A notable challenge for CNS-targeting drugs is delivery, requiring drugs that can efficiently cross the blood-brain barrier. Harnessing safe and practical treatment options, including diet or other microbiome-based approaches, may help accelerate symptomatic relief in PD.

## Materials and methods

**Key resources table**

| Reagent type (species) or resource | Designation | Source or reference | Identifiers | Additional information |
|---|---|---|---|---|
| Cell line (*M. musculus*) | Thy1-α-synuclein (line 61) | *Chesselet et al., 2012*; *Rockenstein et al., 2002* | ASO | |
| Antibody | Anti-beta actin, rabbit polyclonal | Abcam | Cat# ab8227; RRID:AB_2305186 | 1:1,000 |
| Antibody | Anti-aggregated α-synuclein, rabbit polyclonal | Abcam | Cat# ab209538; RRID:AB_2714215 | 1:1000 |
| Antibody | Anti-Iba1, rabbit polyclonal | Wako | Cat# 019–19741; RRID:AB_839504 | 1:1000 |
| Antibody | Anti-tyrosine hydroxylase, chicken polyclonal | Abcam | Cat# ab76442; RRID:AB_1524535 | 1:1000 |
| Antibody | Anti-rabbit IgG-647, donkey polyclonal | Life Technologies | Cat# 1874788; RRID:AB_2536183 | 1:1000 |
| Antibody | Anti-chicken IgG-594, donkey polyclonal | Jackson Immunoresearch | Cat# 703-585-155; RRID:AB_2340377 | 1:600 |
| Antibody | Anti-rabbit IgG, HRP-linked, goat polyclonal | Cell Signaling | Cat# 7074; RRID:AB_2099233 | 1:1000 |
| Antibody | Anti-mouse/human CD11b-APC, rat monoclonal | BioLegend | Cat# 101211; RRID:AB_312794 | 1:1000 |
| Antibody | Anti-mouse CX3CR1-PE/Cyanine7, mouse monoclonal | BioLegend | Cat# 149016; RRID:AB_2565700 | 1:10,000 |
| Antibody | Anti-mouse CD45-Alexa Flour 488, rat monoclonal | BioLegend | Cat# 103121; RRID:AB_493532 | 1:1000 |
| Antibody | Anti-mouse CD16/CD32 Antibody (93), eBioscience (1 mg); rat monoclonal | ThermoFisher | Cat# 14-0161-86; RRID:AB_467135 | 1:100 |
| Antibody | Anti-mouse CD3e Antibody (145–2 C11), PE, eBioscience, hamster monoclonal | ThermoFisher | Cat# 12-0031-82; RRID:AB_465496 | 1:200 |
| Antibody | Anti-mouse CD4 Antibody (GK1.5), APC, eBioscience, rat monoclonal | ThermoFisher | Cat# 17-0041-83; RRID:AB_469321 | 1:200 |
| Antibody | Anti-mouse TCR beta Antibody (H57-597), PerCP-Cyanine5.5, eBioscience, hamster monoclonal | ThermoFisher | Cat# 45-5961-82; RRID:AB_925763 | 1:200 |
| Antibody | Anti-mouse CD8a Antibody (53–6.7), APC-eFluor 780, eBioscience, rat monoclonal | ThermoFisher | Cat# 47-0081-82; RRID:AB_1272185 | 1:200 |
| Antibody | Anti-mouse CD11c Antibody (N418), FITC, eBioscience, hamster monoclonal | ThermoFisher | Cat# 11-0114-82; RRID:AB_464940 | 1:200 |
| Antibody | Anti-mouse CD170 (Siglec F) Monoclonal Antibody (1RNM44N), PE-Cyanine7, eBioscience, rat monoclonal | ThermoFisher | Cat# 25-1702-82; RRID:AB_2802251 | 1:200 |
| Antibody | Anti-mouse Ly-6C Antibody (HK1.4), APC, eBioscience, rat monoclonal | ThermoFisher | Cat# 17-5932-82; RRID:AB_1724153 | 1:200 |

| Reagent type (species) or resource | Designation | Source or reference | Identifiers | Additional information |
|---|---|---|---|---|
| Antibody | Anti-mouse CD103 (Integrin alpha E) Monoclonal Antibody (2E7), PerCP-eFluor 710, eBioscience, hamster monoclonal | ThermoFisher | Cat# 46-1031-82; RRID:AB_2573704 | 1:200 |
| Antibody | Anti-mouse CD64 Antibody (X54-5/7.1), APC-eFluor 780, eBioscience, mouse monoclonal | ThermoFisher | Cat# 47-0641-82; RRID:AB_2735012 | 1:200 |
| Antibody | Anti-mouse CD11b Antibody (M1/70), Super Bright 645, eBioscience, rat monoclonal | ThermoFisher | Cat# 64-0112-82; RRID:AB_2662387 | 1:200 |
| Antibody | APC anti-mouse CD45.2, mouse monoclonal | Tonbo | Cat# 20–0454; RRID:AB_2621576 | 1:200 |
| Antibody | PE-Cy7 anti-mouse Ly6G, rat monoclonal | Tonbo | Cat# 60–1276; RRID:AB_2621860 | 1:200 |
| Antibody | PE-Cy7 anti-mouse TCRb, hamster monoclonal | Tonbo | Cat# 60–5961; RRID:AB_2877098 | 1:200 |
| Antibody | PE-Cy7 anti-mouse/human B220, rat monoclonal | Tonbo | Cat# 60–0452; RRID:AB_2621849 | 1:200 |
| Antibody | FITC anti-mouse CD19, rat monoclonal | Tonbo | Cat# 35–0193; RRID:AB_2621682 | 1:200 |
| Antibody | PE Anti-Mouse MHC Class II (I-A/I-E) (M5/114.15.2), rat monoclonal | Tonbo | Cat# 50–5321; RRID:AB_2621796 | 1:200 |
| Antibody | PE anti-mouse CD115 (CSF-1R) Antibody, rat monoclonal | BioLegend | Cat# 135506; RRID:AB_1937253 | 1:200 |
| Antibody | MHC Class II (I-A/I-E) anti-mouse Antibody (M5/114.15.2), PerCP-eFluor 710, eBioscience, rat monoclonal | ThermoFisher | Cat# 46-5321-82; RRID:AB_1834439 | 1:200 |
| Commercial assay, kit | eBioscience Foxp3 / Transcription Factor Staining Buffer Set | ThermoFisher | Cat# 00-5523-00 | |
| Chemical compound, drug | PLX5622 | DC Chemicals | Cat# DC21518 | |
| Commercial assay, kit | IL-6 Mouse ELISA kit | ThermoFisher | Cat# 88-7064-88 | |
| Commercial assay, kit | TNF-α Mouse ELISA Kit | ThermoFisher | Cat# 88-7324-77 | |
| Commercial assay, kit | Tagment DNA enzyme and buffer kit | Illumina | Cat# 20034197 | |
| Other | Prolong Diamond antifade mountant with DAPI | Invitrogen | Cat# P36971 | |
| Commercial assay, kit | Tissue Extraction Reagent I | ThermoFisher | Cat# FNN0071 | |
| Commercial assay, kit | Chromium Next GEM Single Cell 3' GEM, Library & Gel Bead Kit v3.1 | 10 x Genomics | Cat# 1000128 | |

| Reagent type (species) or resource | Designation | Source or reference | Identifiers | Additional information |
|---|---|---|---|---|
| Commercial assay, kit | Chromium Next GEM Chip G Single Cell Kit | 10 x Genomics | Cat# 1000127 | |
| Commercial assay, kit | Single Index Kit T Set A | 10 x Genomics | Cat# 2000240 | |
| Commercial assay, kit | ChiP DNA clean and concentrator | Zymo | Cat# D5205 | |
| Commercial assay, kit | Direct-zol RNA Microprep | Zymo | Cat# R2062 | |
| Commercial assay, kit | Direct-zol RNA Miniprep | Zymo | Cat# R2050 | |
| Commercial assay, kit | iScript cDNA synthesis kit | Bio-Rad | Cat# 1708890 | |
| Commercial assay, kit | Clarity Western ECL Substrate | Bio-Rad | Cat# 1705060 | |
| Sequence-based reagent | HDAC1_F | This paper | PCR primers | GAACTGCTAAAGTACCACC |
| Sequence-based reagent | HDAC1_R | This paper | PCR primers | CATGACCCGGTCTGTAGTAT-3' |
| Sequence-based reagent | HDAC2_F | This paper | PCR primers | CGGTGTTTGATGGACTCTTTG |
| Sequence-based reagent | HDAC2_R | This paper | PCR primers | CCTGATGCTTCTGACTTCTTG |
| Sequence-based reagent | HDAC6_F | This paper | PCR primers | CTGCATGGCATCGCTGGTA |
| Sequence-based reagent | HDAC6_R | This paper | PCR primers | GCATCAAAGCCAGTGAGATC |
| Sequence-based reagent | HDAC7_F | This paper | PCR primers | CTCGGCTGAGGACCTAGAGA |
| Sequence-based reagent | HDAC7_R | This paper | PCR primers | CAGAGAAATGGAGCCTCTGC |
| Sequence-based reagent | HDAC9_F | This paper | PCR primers | GCGGTCCAGGTTAAAACAGA |
| Sequence-based reagent | HDAC9_R | This paper | PCR primers | GCCACCTCAAACACTCGCTT |
| Sequence-based reagent | GAPDH_F | This paper | PCR primers | ATGGCCTTCCGTGTTCCTA |
| Sequence-based reagent | GAPDH_R | This paper | PCR primers | CCTGCTTCACCACCTTCTTGAT |
| Sequence-based reagent | FFAR2_F | This paper | PCR primers | TTCCCATGGCAGTCACCATC |
| Sequence-based reagent | FFAR2_R | This paper | PCR primers | TGTAGGGTCCAAAGCACACC |
| Sequence-based reagent | FFAR3_F | This paper | PCR primers | ACCGCCGTCAGGAAGAGGGAG |
| Sequence-based reagent | FFAR3_R | This paper | PCR primers | TCCTGCCGTTTCGCSTGGTGG |
| Other | DAPI | Sigma-Aldrich | Cat# 10236276001 | 1:10,000 |
| Other | Aqua Viability Dye | ThermoFisher/Invitrogen | Cat# L34957 | 1:1000 |

## Animals

### Breeding

The Thy1-α-synuclein (ASO; line 61) mouse line was used for all experiments in this study (*Chesselet et al., 2012*; *Rockenstein et al., 2002*). Male BDF1 mice were crossed with female ASO mice expressing the α-synuclein transgene on the X chromosome to generate WT and ASO littermates. Mice were weaned at P21 and housed by genotype on the day of weaning. Male mice were used for all experiments since the human α-synuclein transgene is inserted in the X chromosome, which undergoes random X chromosome inactivation (*Chesselet et al., 2012*).

### Diet experiments

Mice were switched from standard chow to either the cellulose-free control diet or high-fiber prebiotic diet at 5–6 weeks of age and housed in sterile, autoclaved cages with sterile water. Custom fiber mixes were sent from Purdue University for formulation at Envigo Teklad (Madison, WI, USA). After screening diets for efficacy (see *Figure 1—figure supplement 1*), all subsequent studies were performed with Prebiotic diet #1.

PLX5622 was acquired from DC Chemicals and incorporated in the cellulose-free and prebiotic diets at a dosage of 1200 ppm. Mice were switched to the PLX5622 diet at 5–6 weeks of age, and remained on the treatment until 22 weeks of age. Diets were replenished weekly and food intake was measured weekly. Mice were monitored by the lead investigator and Caltech veterinary staff for adverse health effects.

All animal experiments were done under the guidance and approval of Caltech's Institutional Animal Care and Use Committee (IACUC).

## Motor testing

A full battery of motor tests was performed at 22 weeks of age. All motor testing was completed in the same room in a biological safety cabinet between the hours of 6 and 10 of the light phase. Motor testing was completed as described in *Fleming et al., 2004*; *Sampson et al., 2016*. Motor tests were done in the following order: Day 1: beam traversal training, pole training; Day 2: beam traversal training, pole training, wire hang; Day 3: beam traversal test, pole test, hindlimb score, adhesive removal; Day 4: fecal output. Mouse cages were not changed during the duration of testing.

### Beam traversal

Time to cross, errors per step, and number of steps were tested using a plexiglass beam 1 m in length. The beam was constructed of four individual segments, with decreasing width of 1 cm increments along the length of the beam (3.5 cm, 2.5 cm, 1.5 cm, and 0.5 cm). Mice were trained for two consecutive days prior to testing on day 3. On each training day, mice were prompted to cross the beam for three consecutive trials. On testing day mice were recorded using a GoPro camera for analysis of errors per step and number of steps.

### Pole descent

Time to descend a 24-inch pole wrapped in mesh liner was recorded. The pole was placed in the animal's home cage and mice were trained for two consecutive days prior to testing on day 3. Three trials were performed on day 1 of training: trial 1: mice were gently placed head down on the pole 1/3 of the distance from the base, trial 2: mice were placed head down on the pole 2/3 of the distance from the base, trial 3: mice were placed head down on the top of the pole. On day 2 of training, mice were placed on the top of the pole for three consecutive trials. On testing day, mice were placed on the top of the pole for three trials of testing. The timer was stopped once one of the front hindlimbs touched the base of the pole. Time to descend was averaged across all three trials.

### Adhesive removal

A 0.25 in. adhesive, round sticker was placed on the nose of the mouse. The mouse was subsequently placed in its home cage (without cagemates) and time to remove the adhesive was recorded. Time to remove was averaged across two trials.

## Wire hang

Mice were placed in the middle of a rectangular wire grid placed over a sterile, clean cage. the wire grid was gently inverted with the mouse hanging over the cage. Time to fall was recorded as the time between grid inversion and the mouse falling off the grid. Maximum time was set to 60 s. Time to fall was averaged across two trials.

## Hindlimb score

Mice were gently held upwards in the air by the mid-section of their tail and hindlimb movement was observed. Mice were given a score of 0, 1, 2, or 3 depending on the movement and flexibility of their rear hindlimbs. The score was assessed by two experimenters and the average score was reported.
    Scores were assigned as follows:

> 0: rear hindlimbs were flexible and mobile, with a complete range of motion; no inward clasping was observed
> 1: rear hindlimbs exhibited mild rigidity with hindlimbs orienting inward slightly
> 2: rear hindlimbs oriented inward, but were not completely clasped
> 3: rear hindlimbs were firmly clasped together

## Microglia isolation and sequencing

### Microglia isolation

Microglia were isolated from mouse brains at 22 weeks of age. For all experiments, samples were pooled from 4 to 6 mice/treatment group. Mice were anesthetized and perfused with ice-cold PBS. Brain regions of interest were dissected and homogenized using mechanical dissociation. Single cell suspensions were obtained using a Dounce homogenizer. A 37/70 Percoll density gradient was used to separate cells from debris and myelin. Following Percoll separation, cells were washed and stained with Cd11B (1:1000, Biolegend), CX3CR1 (1:10,000, Biolegend), CD45 (1:1000, Biolegend), and DAPI (1:10,000, Sigma-Aldrich). All steps were performed in microglia staining buffer (1 X HBSS, 1% BSA and 1 mM EDTA). Cells were sorted in a FACSAria III Fusion flow cytometer (BD Biosciences). Live CD11b+, CX3CR1+, and CD45 (low) cells were identified as microglia and collected for analysis. The full protocol can be found at protocols.io (https://doi.org/10.17504/protocols.io.kqdg3p7bel25/v1).

### Single-cell sequencing

The v3.1 Chromium Next GEM single cell reagent kit from 10 x genomics was used for scRNAseq of FACS-purified microglia. Between 2 and 4000 cells were loaded on the Next GEM chip for substantia nigra samples, with 1000–1700 cells/group recovered for analysis. For striatum samples, approximately 8–16,000 cells were loaded on the Next GEM chip, with 5–10,000 cells/group recovered for analysis. Library construction was completed according to the manufacturer's instructions. Samples were tagged with a unique sample index, pooled, and sequenced with an average depth of 111 k reads/cell on a NovaSeq 6000 sequencing platform (Illumina). Cell Ranger software (10 X Genomics) was used for sequence alignment, cluster analysis, and identification of differentially expressed genes between groups. ShinyGO was used for gene ontology and pathway analysis (*Ge et al., 2020*).

### Single-cell transcriptomic analysis

The data were first filtered by removing cells with less than 200 genes and genes that were expressed in less than 100 cells. Gene counts were normalized by dividing the number of times a particular gene appeared in a cell (gene cell count) by the total gene counts in that cell. Counts were multiplied by a constant factor (5000), a constant value of 1 was added to avoid zeros, and then the data were log transformed. Data analysis steps including Leiden clustering, differential gene expression analysis, and plotting of marker genes were performed using the Scanpy package (*Wolf et al., 2018*).

### ATAC Seq

FACS-purified microglia were collected as described above and resuspended in 50 µL of ice-cold lysis buffer (10 mM Tris-HCl, pH 7.4, 10 mM NaCl, 3 mM MgCl$_2$, 0.1% IGEPAL CA-630). Cells were

spun down at 500 xg for 10 min at 4 °C. Supernatant was discarded and a transposition reaction was performed on the cell pellet using the Illumina Tagment DNA enzyme and buffer kit. Samples were purified using the Zymo ChIP DNA clean and concentrator kit and transposed DNA was eluted in elution buffer. Two independent trials were completed for the experiment.

## Immunohistochemistry
### Sectioning

Twenty-two-week-old WT and ASO mice were anesthetized with pentobarbital (Euthasol). Mice were perfused with ice-cold phosphate buffered saline (PBS) and 4% paraformaldehyde (PFA). Brains were removed and placed in tissue culture plates with 4% PFA for 48 hr before transfer to PBS +0.05% sodium azide. Whole brains were embedded in agarose and sliced coronally into 50 µM sections using a vibratome. Free-floating sections were placed in PBS +0.05% sodium azide and stored at 4 °C until staining.

### Staining
Sections were permeabilized for 30 min in 3% BSA, 0.5% Triton X-100 in PBS, blocked for 1 hr in 3% BSA, 0.1% Triton X-100 in PBS, and stained with IBA-1 (1:1000, Wako, anti-rabbit) and tyrosine hydroxylase (Th) (1:1000, Abcam, anti-chicken) overnight at 4 °C (protocol adapted from *Datta et al., 2018*). Sections were then stained with anti-rabbit IgG AF-647 (1:1,000, Life Technologies) and anti-chicken IgG AF-594 (1:600, Jackson ImmunoResearch). Slices containing brain regions of interest were mounted on a cover slip using ProLong Diamond anti fade mountant with 4',6-diamidino-2-phenylindole (DAPI). Coverslips were stored at 4 °C until imaging.

### Imaging
Images were obtained on a Zeiss LSM800. For diameter measurements: images were taken with a 20 X objective, with 3 pictures taken per brain region of interest. Imaris Software was used to measure the diameters of cells, with 30–70 cells counted per brain region/animal. For 3D reconstruction: z-stack images were taken with 1.00 µm steps in the z-direction with a 40 X objective. 3D reconstruction was done in the Imaris Software, with 3–6 cells analyzed per brain region/animal.

## α-Synuclein aggregation assays
Susbtantia nigra and striatum were dissected on ice from 22-week-old mice and stored at –80 °C until used.

### Protein extraction
Brain tissues were lysed using Tissue Extraction Reagent (Thermo Fisher) and protease inhibitor. Samples were homogenized for 90 s using a bead beater and were placed directly on ice for 10–15 min following homogenization. Lysates were centrifuged at 10 k rpm for 5 min and supernatants were collected and stored at –80 °C for later use. The full protocol can be found at protocols.io (https://doi.org/10.17504/protocols.io.5jyl896o6v2w/v1).

### α-Synuclein aggregation
Levels of aggregated α-synuclein were determined using the dot blot assay. Samples were quantified using the Pierce BCA Protein assay kit (Thermo Fisher) and normalized to equal concentrations between 0.5–1.0 ng/µL in water. One µg of sample was spotted on dry nitrocellulose membrane (0.45 µm). Samples were blocked in 5% skim milk in Tris-buffered saline with 0.1% Tween-20 (TBS-T) and stained with anti-aggregated α-synuclein antibody (1:1000, Abcam) overnight at 4 °C. The next day, blots were stained with anti-rabbit IgG-HRP (1:1000, Cell Signaling) for 2 hr. Signal was detected using Clarity chemiluminescence substrate (Bio-Rad) and imaged on a Bio-Rad digital imager. Integrated density is reported as the intensity of an identically-sized area of each dot for each sample. The full protocol can be found at protocols.io (https://doi.org/10.17504/protocols.io.261gen2xdg47/v1).

## RNA extraction and qPCR

Brain regions were dissected on ice from 22-week-old mice and stored at –80 °C in RNAlater solution (Thermo Fisher) until RNA extraction.

### RNA extraction

RNA was extracted using either Direct-zol RNA Microprep or Miniprep kit (Zymo Research) depending on the size of the brain region. qPCR: RNA was transcribed using the iScript cDNA synthesis kit (Bio-Rad) per the manufacturer's instructions. SYBR Green master mix was used for qPCR reactions. Primers used for experiments were: HDAC1: 5'-GAACTGCTAAAGTACCACC-3' & 5'-CATGACCC GGTCTGTAGTAT-3; HDAC2: 5'-CGGTGTTTGATGGACTCTTTG-3' & 5'-CCTGATGCTTCTGACTTCTT G-3'; HDAC6: 5'-CTGCATGGCATCGCTGGTA-3' & 5'-GCATCAAAGCCAGTGAGATC-3'; HDAC7: 5'-CTCGGCTGAGGACCTAGAGA-3' & 5'-CAGAGAAATGGAGCCTCTGC-3'; HDAC9: 5'-GCGGTCCA GGTTAAAACAGAA-3' & 5'-GCCACCTCAAACACTCGCTT-3'; GAPDH: 5'-CATGGCCTTCCGTGTT CCTA-3' & 5'- CCTGCTTCACCACCTTCTTGAT-3'; FFAR2: 5'-TTCCCATGGCAGTCACCATC-3' & 5'-TGTAGGGTCCAAAGCACACC-3'; FFAR3: 5'-ACCGCCGTCAGGAAGAGGGAG-3' & 5'TCCTGCCG TTTCGCSTGGTGG-3'.

## Isolation of immune cells from intestinal lamina propria/spleen and flow cytometry

For isolation of intestinal lamina propria cells, the small and large intestines were dissected and placed immediately into ice-cold PBS. After mesenteric fat and Peyer's patches (small intestine) were removed, the intestines were longitudinally opened and luminal contents were washed out with cold PBS. Tissue pieces were washed for 10 min in 1 mM dithiothreitol (DTT)/PBS at room temperature on a rocker to remove mucus, followed by a wash for 25 min in 10 mM EDTA/30 mM HEPES/PBS at 37 °C on a platform shaker (180 rpm) to remove epithelium. After a 2 min wash in complete RPMI, tissue was digested in a six-well plate for 1.5 hr in complete RPMI with 150 U/mL (small intestine) or 300 U/mL (large intestine) collagenase VIII (Sigma-Aldrich) and 150 µg/mL DNase (Sigma-Aldrich) in a cell culture incubator (5% $CO_2$). Tissue digests were passed through a 100 µm cell strainer and separated by centrifugation (1200 xg for 20 min) using a 40/80% Percoll gradient. Immune cells were collected at the 40/80% interface. For the spleen, the tissue was passed through a 100 µm cell strainer and incubated in red cell lysis buffer (Sigma-Aldrich) for 8 min at room temperature. Both spleen and intestine immune cells were washed with 0.5% BSA/PBS before staining and fixation (eBioscience Foxp3 / Transcription Factor Staining Buffer Set).

For flow cytometry staining, CD16/32 antibody (eBioscience) was used to block non-specific binding to Fc receptors before surface staining. Immune cells were stained with antibodies against the following markers: CD103 (PerCP-efluor710), CD11b (SuperBright645), CD11c (FITC), CD19 (FITC), CD3e (PE), CD4 (APC), CD45.2 (BV421), CD64 (APC-Cy7), CD8a (APC-e780), CSF1R (PE), Ly6C (APC), MHCII I-A/I-E (PE or PerCP-efluor710), TCRβ (PerCP-Cy5.5). For some panels, a lineage marker mix (Lin) contained TCRβ, B220, Ly6G and Siglec-F (PE-Cy7). Live and dead cells were discriminated by Live/Dead Fixable Aqua Dead Cell Stain Kit (Invitrogen).

## Gut microbiome profiling

### Metagenomic sequencing

Shotgun sequencing libraries were generated using the Kapa HyperPlus protocol on gDNA extracted from mouse fecal pellets. Samples were sequenced using 150 bp paired end reads on an Illumina NovaSeq 6000 at the UCSD IGM Genomics Center.

### Metagenomic analyses

Quality control filtering and read alignment of metagenomic reads was conducted with Qiita (study-id 13244). First, adapter removal and quality trimming were conducted using Atropos v1.1.24. To generate taxonomic and functional gene-level profiles we applied the Woltka v0.1.1 pipeline to align reads against the Web of Life database (*Zhu et al., 2019*) using Bowtie2 v2.3.0 (*Langmead and Salzberg, 2012*), followed by generation of Operational Genomic Units (*Zhu et al., 2021*). Downstream

statistical analyses and data visualization was conducted in R (v4.1.0). For community-level measures, including alpha- and beta-diversity, Woltka-generated taxonomic predictions at the species level were rarefied to an even depth of 321,980 counts. Alpha-diversity metrics including Observed Species, Simpson's Evenness, and Gini's Dominance were calculated using the microbiome R package and tested for statistical significance using a one-way ANOVA for treatment group and post-hoc Tukey's test for pairwise comparisons. Assessment of between-sample diversity was accomplished using the Bray-Curtis distance. We estimated metadata-explained variance using the Bray-Curtis distance with permutational multivariate analysis of variance (PERMANOVA) with 9,999 permutations followed by multiple hypothesis testing corrections using the Benjamini-Hochberg method (FDR = 0.1). Differential abundance testing was conducted using Multivariable Association with Linear Models (MaAsLin2) (*Mallick et al., 2021*). For data preparation, we applied a 10% prevalence filter, total sum scale normalization, and an arcsine square root transformation for variance stability. We then applied a feature-level-specific variance filter based on the variance distribution and the number of features present at each level. MaAsLin2 linear models were fit with genotype and diet variables as fixed effects.

## SCFA fecal measurements (LC-MS)

Fecal samples were collected from mice at 22 weeks of age and stored at –80 °C until analysis. Sample preparation: Mouse fecal samples were extracted and derivatized as described previously (*Chan et al., 2017*). Briefly, ice-cold extraction solvent (1:1 v/v acetonitrile/water) was added to fecal sample at a ratio of 2 µL:1 mg sample and internal standard mix to a final concentration of 100 µM. The suspension was vortex mixed for 3 min at room temperature, sonicated for 15 min, and then centrifuged at 18,000 x g for 15 min at 4 °C. An aliquot of 100 µL was subsequently derivatized using a final concentration of 10 mM aniline and 5 mM 1-ethyl-3-(3-dimethylaminopropyl)carbodiimide hydrochloride (EDC) (ThermoFisher) for 2 hr at 4 °C. The derivatization reaction was quenched using a final concentration of 18 mM succinic acid and 4.6 mM 2-mercaptoethanol for 2 hr at 4 °C. All samples were stored at 4 °C until analysis on the same day. Mixed calibrators of acetic acid, propanoic acid, butyric acid and isobutyric acid (10 nM - $10\times10^3$ nM) (Sigma-Aldrich) together with single- and double- blanks, spiked with internal standard mix (acetic acid-d3, propanoic acid-d2, butyric acid-d2) (Pointe-Claire) to a final concentration of 100 µM were prepared and subjected to the same sample preparation procedure as fecal samples. The full protocol can be found at protocols.io (https://doi.org/10.17504/protocols.io.bp2l61rrkvqe/v1).

## Liquid chromatography mass spectrometry (LC-MS)

Derivatized samples were analyzed using an ultra-high-performance liquid chromatography (UHPLC) system 1290 connected to a quadrupole time of flight (Q-TOF 6545) mass spectrometer (Agilent Technologies) equipped with an orthogonal DUAL AJS-ESI interface. Samples were subjected to reverse phase C18 separation (Phenomenex Scherzo SS-C18 100x2 mm) and data were collected in positive ion mode. Data were acquired from 50 to 750 m/z-1 at 2 spectra $s^{-1}$. Electrospray ionization (ESI) source conditions were set as follows: gas temperature 325 °C, drying gas 9 L $min^{-1}$, nebulizer 35 psi, fragmentor 125 V, sheath gas temperature 350 °C, sheath gas flow 8 L $min^{-1}$, nozzle voltage 1000 V. For reverse phase C18 chromatographic separation, a two-solvent gradient running at 0.3 mL $min^{-1}$ (Mobile Phase: A: 100:0.1 Water:Formic Acid, B: 100:0.1 Isopropanol:Formic Acid) was used. The column was equilibrated at 15% B for 1 min and a sample was introduced. The solvent ratio was then increased from 15% B to 100% B over 13 min and then reduced back to 15% B over 2 min. Injection volume was 5 µL with a column temperature of 45 ° C. The LC-MS/MS data acquired using Agilent Mass Hunter Workstation (.d files) were processed in quantitative analysis software (Agilent Technologies) for quantitative analysis of samples. The linear calibration plots for acetic acid, propanoic acid, butyric acid and isobutyric acid were constructed using peak area ratios of each analyte to the IS versus the concentrations of calibrators (x) with 1 /x weighting, and the least squares linear regression equations were obtained as the calibration equations for individual analytes.

## SCFA brain measurements (UHP-LC)

Striatum and substantia nigra were dissected from 22-week-old mice, placed in dry ice, and stored at –80 °C until analysis. Samples were analyzed by BIOTOOLS CO. using an ultra-high-performance liquid chromatography (UHPLC) system. Brain tissue samples were extracted with 70% methanol for

30 min, using a sample:solvent ratio of 1 mg:40 µL. The sample was centrifuged at 21,380 rcf for 5 min at 4 °C. The supernatant was used for derivatization procedures. Each sample was mixed with 5 µL of 0.1 mM internal standard and 200 µL each of pyridine, 1-EDC-HCl, and 2-NPH-HCl solutions as reaction-assistive agents, and reacted at 45 °C for 20 min. 100 µL of potassium hydroxide solution was added (to stop the reaction) and reacted at 45 °C for 15 min. After cooling, the mixture was ultrasonicated with 1 mL of phosphoric acid aqueous solution and 2 mL of ether for 3 min and then centrifuged for 5 min at 2,054 rcf. The ether layer was collected and spun-dry. The sample was reconstituted with 25 µl MeOH. Mass analysis: Each sample (2 µL) was injected into a Vanquish ultra-high-performance liquid chromatography (UHPLC) system coupled with SCIEX QTrap 5500. UHPLC parameters were set as follows: A CSH 1.7 µm, 2.1x100 mm column (Waters) was used. The column oven temperature was set at 45 °C. The binary mobile phase included deionized water containing 5 mM ammonium acetate as solvent A, and acetonitrile with 5 mM ammonium acetate as solvent B. The flow rate was 0.35 mL/min with a linear gradient elution over 15 min. Reagent 1: Pyridine (Sigma-Aldrich) was adjusted with methanol to 3% (V/V) (*Weng et al., 2020*).

## Statistical analysis

Graphpad Prism software (version 9.0) was used for statistical analysis. Data presented represent mean ± SEM, with each data point representing values from an individual mouse. All behavioral and molecular data were analyzed by two-way ANOVA followed by Tukey's multiple comparisons test, unless stated otherwise. $*p < 0.05$, $**p < 0.01$, $***p < 0.001$, and $****p < 0.0001$.

## Acknowledgements

We thank members of the Mazmanian laboratory and Dr. Catherine Oikonomou for critical review of this manuscript. We thank the Caltech Office of Laboratory Animal Research (OLAR) for animal husbandry, Dr. Wei-Li Wu for assistance with SCFA brain measurements, Dr. Sisi Chen and the Caltech Single-Cell Profiling and Engineering Center (SPEC) for technical assistance and support, the Caltech Flow Cytometry and Cell Sorting Facility for technical assistance, the Caltech Bioinformatics Center for data analysis support, and the Caltech Biological Imaging Facility (BIF) for training and use of confocal microscopes. We thank Prof. Chen-Chih Hsu's laboratory in the Department of Chemistry at National Taiwan University and BIOTOOLS Co, Ltd. for the feces and brain SCFA measurements. RA was supported by the U.S. Department of Defense, the Donna and Benjamin M Rosen Bioengineering Center, and the Biotechnology Leadership Program at Caltech. This study was funded by grants to SKM from the U.S. Department of Defense (PD160030), Heritage Medical Research Institute (HMRI-15-09-01), and by the joint efforts of the Michael J Fox Foundation for Parkinson's Research (MJFF) and the Aligning Science Across Parkinson's (ASAP) initiative. MJFF administers the grant (ASAP-000375) on behalf of ASAP and itself.

## Additional information

### Competing interests

Thaisa M Cantu-Jungles, Ali Keshavarzian, Bruce R Hamaker: has equity in RiteCarbs, a company developing prebiotic diets for Parkinson's disease. Sarkis K Mazmanian: has equity in Axial Therapeutics, a company developing gut-restricted drugs for Parkinson's disease. The other authors declare that no competing interests exist.

### Funding

| Funder | Grant reference number | Author |
|---|---|---|
| U.S. Department of Defense | PD160030 | Sarkis K Mazmanian |
| Heritage Medical Research Institute | HMRI-15-09-01 | Sarkis K Mazmanian |

| Funder | Grant reference number | Author |
| --- | --- | --- |
| Aligning Science Across Parkinson's | ASAP-000375 | Sarkis K Mazmanian |

The funders had no role in study design, data collection and interpretation, or the decision to submit the work for publication.

## Author contributions

Reem Abdel-Haq, Conceptualization, Data curation, Formal analysis, Validation, Investigation, Visualization, Methodology, Writing – original draft, Writing – review and editing; Johannes CM Schlachetzki, Formal analysis, Investigation, Visualization, Writing – review and editing; Joseph C Boktor, Formal analysis, Visualization, Writing – review and editing; Thaisa M Cantu-Jungles, Investigation, Writing – review and editing; Taren Thron, Investigation; Mengying Zhang, Investigation, Methodology; John W Bostick, Livia H Morais, Data curation, Investigation; Tahmineh Khazaei, Methodology; Sujatha Chilakala, Data curation, Formal analysis, Methodology; Greg Humphrey, Data curation; Ali Keshavarzian, Resources, Funding acquisition; Jonathan E Katz, Data curation, Funding acquisition, Methodology; Matthew Thomson, Formal analysis, Methodology; Rob Knight, Data curation, Supervision, Methodology; Viviana Gradinaru, Data curation, Supervision; Bruce R Hamaker, Resources, Data curation, Supervision, Funding acquisition, Methodology; Christopher K Glass, Conceptualization, Supervision, Methodology; Sarkis K Mazmanian, Conceptualization, Supervision, Funding acquisition, Writing – review and editing

## Author ORCIDs

Reem Abdel-Haq http://orcid.org/0000-0002-7418-5736
Livia H Morais http://orcid.org/0000-0002-5738-2658
Ali Keshavarzian http://orcid.org/0000-0002-7969-3369
Viviana Gradinaru http://orcid.org/0000-0001-5868-348X
Christopher K Glass http://orcid.org/0000-0003-4344-3592
Sarkis K Mazmanian http://orcid.org/0000-0003-2713-1513

## Ethics

All animal experiments were done under the guidance and approval of Caltech's Institutional Animal Care and Use Committee (IACUC).

## Decision letter and Author response

Decision letter https://doi.org/10.7554/eLife.81453.sa1
Author response https://doi.org/10.7554/eLife.81453.sa2

# Additional files

## Supplementary files

• Supplementary file 1. Composition of custom-made prebiotic diets.

• Supplementary file 2. Differentially expressed genes (DEGs) in microglia of the Substantia Nigra and Striatum.
 (a) DEGs in Control-WT vs Control-ASO microglia in the Substantia Nigra. Log fold change relative to Control-WT. (b) DEGs in Control-ASO vs Prebiotic-ASO microglia in the Substantia Nigra. Log fold change relative to Control-ASO. (c) DEGs in Control-WT vs Control-ASO microglia in the Striatum. Log fold change relative to Control-WT. (d) DEGs in Control-ASO vs Prebiotic-ASO microglia in the Striatum. Log fold change relative to Control-ASO.

• Supplementary file 3. Genes detected in microglia in scRNA-seq in the Substantia Nigra and Striatum.

• MDAR checklist

## Data availability

All datasets generated or analyzed in this study can be found through the Zenodo depository: https://doi.org/10.5281/zenodo.6377704. All experimental protocols can be found on protocols. io.

The following dataset was generated:

| Author(s) | Year | Dataset title | Dataset URL | Database and Identifier |
|---|---|---|---|---|
| Abdel-Haq R, Mazmanian S, Schlachetzki JCM, Boktor JC, Cantu-Jungles TM, Thron T, Zhang M, Bostick JW, Khazaei T, Chilakala S, Morais LH, Humphrey G, Keshavarzian A, Hamaker BR, Katz JE, Thomson M, Knight R, Glass CK, Gradinaru V | 2022 | A prebiotic diet modulates microglia response and motor deficits in α-synuclein overexpressing mice | https://doi.org/10.5281/zenodo.6377704 | Zenodo, 10.5281/zenodo.6377704 |

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
