## [Editor Report]

What should Parkinson's Disease patients eat? This study shows that dietary fiber impacts gut microbes and immune cells in the brain of a mouse model of Parkinson's. These findings will enable follow-up studies aimed at figuring out how this works and efforts to translate these findings to improve patient care.

---

## [Decision Letter]

**Decision letter after peer review:**

Thank you for submitting your article "A prebiotic diet modulates microglial states and motor deficits in α-synuclein overexpressing mice" for consideration by *eLife*. Your article has been reviewed by 3 peer reviewers, including Peter J Turnbaugh as Reviewing Editors and Reviewer #3, and the evaluation has been overseen by Carla Rothlin as the Senior Editor. The following individual involved in the review of your submission has agreed to reveal their identity: Martin Valdearcos (Reviewer #1).

Essential revisions:

1) Please edit the main text to clarify and discuss the points listed below.

*Reviewer #1 (Recommendations for the authors):*

In Figure 5, the representative images showing Iba 1+ cells for each condition seem to have different magnifications.

*Reviewer #2 (Recommendations for the authors):*

Results section

Line 118: "We fed each of the three prebiotic diets (prebiotic #1, #2, #3) to male ASO mice from 5-22 weeks of age". Is there a reason why this study was done with male animals only?

Line 132: "Interestingly, concentrations of propionate, butyrate, and isobutyrate were not significantly different between wild type (WT) and ASO mice fed a control diet (Figure 1E)" What makes this finding "interesting"?

Line 143: "We speculate that this difference may be attributable to regional differences in microglia density, gene expression, and clearance activity, with the SN having a relatively higher density of microglia (Grabert et al. 2016; Y.-L. Tan, Yuan, and Tian 6 145 2020)". This sentence could be written differently to avoid mentioning that a "difference" is attributable to "regional differences", and the repeat of microglia density. Furthermore, rather than citing studies to support the "speculation" of the potential importance of differences in microglia density, would it be possible to localize and count the number of microglial cells in brain sections of their various mouse populations, as described in lines 177-179?

Line 174: "In ASO mice, microglia reactivity in response to αSyn overexpression appears at 4-5 weeks of age in the STR and at 20-24 weeks of age in the SN (Watson et al. 2012)". The term "microglia reactivity" is imprecise. Can the authors indicate what is measured to determine the reactivity of microglia?

Considering the finding that the depletion of microglial cells in ASO mice decreased the beneficial effect of the prebiotic treatment on motor impairment, it would have been interesting to investigate whether any changes occurred in the dopaminergic system in the SN and STR, including the number of DN neuronse and the extent of their arborization when microglial cells were depleted.

Discussion

Line 355: "Microglia have been increasing linked to neurodegenerative disorders…" Please modify "increasing linked".

Materials and methods

Line 550: Please indicate that the diet and PLX5622 treatment continued until the sacrifice of the animals at week-22.

*Reviewer #3 (Recommendations for the authors):*

Lines 169-171: Over-interpretation. It is dangerous to attempt to infer immune effects based on taxonomic abundances.

Figure 1: Should specify which prebiotic is used in the panels and legend.

Figure 2G – colors are hard to distinguish, especially Bacteroidetes and Verrucomicrobia.

---

## [Author Response]

Reviewer #1 (Recommendations for the authors):In Figure 5, the representative images showing Iba 1+ cells for each condition seem to have different magnifications.

The two images in Figure 5C are the same magnification (20x), but appear somewhat different as the microglia in the PLX5622-treated group are fewer and less crowded compared to the untreated group.

Overall, we thank the referee for their time in reviewing our manuscript and their very helpful suggestions.

Reviewer #2 (Recommendations for the authors):Results sectionLine 118: "We fed each of the three prebiotic diets (prebiotic #1, #2, #3) to male ASO mice from 5-22 weeks of age". Is there a reason why this study was done with male animals only?

We thank the referee for raising this point. Female a-synuclein over-expressing (ASO) ‘Line 61’ mice express lower levels of human α-synuclein because the transgene is inserted in the X chromosome, which undergoes random chromosome inactivation that introduces variability in expression levels. This issue has been well documented in the literature and male ASO mice are routinely used to model aspects of PD (PMID: 22350713). To clarify for the readers of the current manuscript, we have added the following sentence to the Methods section:

“Male mice were used for all experiments since the human α-synuclein transgene is inserted in the X chromosome, which undergoes random X chromosome inactivation (Chesselet et al., 2012)”

Line 132: "Interestingly, concentrations of propionate, butyrate, and isobutyrate were not significantly different between wild type (WT) and ASO mice fed a control diet (Figure 1E)" What makes this finding "interesting"?

We describe this finding as “interesting” since SCFA levels have been reported to be lower in PD compared to health subjects. In retrospect, we agree that extrapolation between mouse models and humans may be a minor or unnecessary point, and have removed the word. We thank the referee for the helpful feedback.

Line 143: "We speculate that this difference may be attributable to regional differences in microglia density, gene expression, and clearance activity, with the SN having a relatively higher density of microglia (Grabert et al. 2016; Y.-L. Tan, Yuan, and Tian 6 145 2020)". This sentence could be written differently to avoid mentioning that a "difference" is attributable to "regional differences", and the repeat of microglia density. Furthermore, rather than citing studies to support the "speculation" of the potential importance of differences in microglia density, would it be possible to localize and count the number of microglial cells in brain sections of their various mouse populations, as described in lines 177-179?

In this study we are primarily interested in characterizing how diet-induced changes to the microbiome alter microglia cell activity, specifically looking at changes between groups. The higher density of microglia in the SN has been previously documented and additionally, we do not observe any differences in microglia density between control and prebiotic-fed mice. To directly address this concern, we have now removed the phrase “with the SN having a relatively higher density of microglia” to reduce emphasis on the differences in microglia numbers between brain regions, and therefore reduce speculation. We thank the referee for this important comment.

Line 174: "In ASO mice, microglia reactivity in response to αSyn overexpression appears at 4-5 weeks of age in the STR and at 20-24 weeks of age in the SN (Watson et al. 2012)". The term "microglia reactivity" is imprecise. Can the authors indicate what is measured to determine the reactivity of microglia?

There have been ongoing discussions in the microglia research community to more precisely characterize microglia physiological states (http://dx.doi.org/10.2139/ssrn.4065080). Rather than use the term ‘activation’, which can encompass both pro- and anti-inflammatory pathways, we describe microglial reactivity in response to the presence of a-synuclein aggregates. There are several ways to measure microglial reactivity. In our study we measured microglia reactivity using imaging to look at microglia morphology and scRNAseq to investigate gene expression profile. The morphology of microglia can indicate whether microglia are resting vs. reactive, as was first shown in Watson et al., for ASO mice. Reactive microglia have an ameboid morphology characterized by a larger cell body diameter and fewer processes. Reactive microglia also upregulate a variety of immune markers depending on the specific stimulus. Our scRNAseq experiment showed reactive microglia upregulate pro-inflammatory markers and pathways in ASO mice. Therefore, we believe that our data clearly show changes in microglial reactivity using standard methaos, and that ‘reactivity’ is the proper term based on the literature. We thank the referee for this comment.

Considering the finding that the depletion of microglial cells in ASO mice decreased the beneficial effect of the prebiotic treatment on motor impairment, it would have been interesting to investigate whether any changes occurred in the dopaminergic system in the SN and STR, including the number of DN neuronse and the extent of their arborization when microglial cells were depleted.

This is a very good point. It is well established that ASO mice do not show numerical changes in dopaminergic or other neuronal populations in the SN or STR at the 22-week age that we report in this study (PMIDs: 22350713, 21488084), therefore there is no phenotype to investigate. Answering this question would require aging mice beyond 1 year of age, when they first start showing loss of DA neurons (PMID: 21488084). We thank the referee for this comment.

DiscussionLine 355: "Microglia have been increasing linked to neurodegenerative disorders…" Please modify "increasing linked".

We have corrected the phrase to “increasingly linked” and thank the referee for observing this text error.

Materials and methodsLine 550: Please indicate that the diet and PLX5622 treatment continued until the sacrifice of the animals at week-22.

We agree, and have updated the text to now read:

“Mice were switched to the PLX5622 diet at 5-6 weeks of age, and remained on the treatment until 22 weeks of age.”

Reviewer #3 (Recommendations for the authors):Lines 169-171: Over-interpretation. It is dangerous to attempt to infer immune effects based on taxonomic abundances.

We agree and have reworded the sentence to read:

"Overall, feeding of a prebiotic diet appears to qualitatively restructure the ASO microbiome toward increased relative abundances of taxa associated with potentially protective effects."

Further, to address the point above about caution on specific dietary recommendations, we have removed the term “therapeutic” from the final sentence of the manuscript.

Figure 1: Should specify which prebiotic is used in the panels and legend.

We have added the term “Prebiotic #1” to the Figure 1 legend to clarify this point, and added the following phrase to the Methods section:

“After screening diets for efficacy (see Figure 1—figure supplement 2), all subsequent studies were performed with Prebiotic diet #1.”

Figure 2G – colors are hard to distinguish, especially Bacteroidetes and Verrucomicrobia.

We agree and have changed the coloring in Figure 2G to increase visual clarity.